# Do Comorbidities and Daily Medication before SARS-CoV-2 Infection Play a Role in Self-Reported Post-Infection Symptoms?

**DOI:** 10.3390/jcm11216278

**Published:** 2022-10-25

**Authors:** Dovilė Važgėlienė, Raimondas Kubilius, Indre Bileviciute-Ljungar

**Affiliations:** 1Department of Physical Rehabilitation Medicine, Kaunas Clinic of the Lithuanian University of Health Sciences, 50161 Kaunas, Lithuania; 2Department of Nursing, Lithuanian University of Health Sciences, 44307 Kaunas, Lithuania; 3Department of Clinical Sciences, Karolinska Institute at Danderyd University Hospital, 182 57 Stockholm, Sweden; 4Multidisciplinary Pain Clinic, St. Göran Hospital, 112 19 Stockholm, Sweden

**Keywords:** SARS-CoV-2 virus, medication, comorbidities, post-COVID-19 condition

## Abstract

This study investigated the associations between health status before SARS-CoV-2 infection and persistent symptoms after acute infection. Data were collected from participants older than 18 years and more than 28 days after acute SARS-CoV-2 infection using an online survey. Sociodemographic data, comorbidities, and daily medication before infection, as well as acute and persistent symptoms were analysed. Among the 1050 participants (mean age 41 years, 88% women, 56% with higher education, 93% working), 538 (51%) reported being healthy and 762 (73%) reported not taking any daily medication prior to infection. Positive laboratory testing was reported by 965 (92%) participants; asymptomatic infection was reported by 30 (3%); and 999 (95%) stayed at home during their acute infection. Reduced physical capacity (40%), fatigue (39%), cognitive difficulties (30–34%), altered sense of smell (24%), headache (20%), tachycardia (20%), unstable mood (19%), hair loss (17%), and insomnia (17%) were the most often reported symptoms. Those taking daily medication before infection reported increased frequency of both acute and persistent symptoms, except for decreased frequency of persistent altered smell and taste. The presence of persistent symptoms was predicted by taking daily medication before infection and by the total number of acute symptoms. Comorbidities before infection did not predict persistent symptoms. Therefore, the role of medication needs further investigation in both acute SARS-CoV-2 infection and post-COVID-19 condition.

## 1. Introduction

Since December of 2019, the COVID-19 pandemic has affected several hundred million people all over the world, resulting in millions of deaths and overloaded healthcare systems in many countries. The World Health Organization (WHO) has also alerted healthcare providers regarding post-COVID-19 condition and estimates that approximately 10% of all infected people may suffer from it [1]. According to the WHO, post-COVID-19 condition occurs in individuals with a history of probable or confirmed SARS-CoV-2 infection, usually starting 3 months from the onset of COVID-19, with symptoms that last for at least 2 months and cannot be explained by an alternative diagnosis. Common symptoms include fatigue, shortness of breath, cognitive dysfunction, and others, and these generally have at least some impact on everyday functioning. Symptoms may be new onset, may follow initial recovery from an acute COVID-19 episode, or may persist from the initial illness. Symptoms may also fluctuate or relapse over time [1]. 

It is estimated that the incidence of persistent symptoms in non-hospitalized patients is between 22% and 38% [2], while in hospitalized patients it is up to 85% [3]. It might be speculated that post-COVID-19 condition can have different pathways in hospitalized and non-hospitalized patients because older male patients dominate among hospitalized patients [4,5], while middle-aged women more often report persistent symptoms, in survey studies [6]. Fatigue has been found to be the most common symptom in post-COVID-19 condition independent of disease severity [7]. The role of comorbidities has been studied and reported in hospitalized patients, showing an increased risk of mortality and increased need for intensive care during acute SARS-CoV-2 infection in those with cardiovascular, pulmonary, metabolic, renal, and other somatic diseases [8,9,10]. 

The predictors of post-COVID-19 condition have been studied in hospitalized patients, reporting that comorbidities might predict persistent symptoms in both inpatients and outpatients [11], while others did not find clinical predictors for persistent symptoms in hospitalized patients [12]. A recent survey study with several thousand participants reported that acute symptomatology might predict post-COVID-19 condition [13], and increasing age, female sex, white ethnicity, poor pre-pandemic general and mental health, overweight/obesity, and asthma were associated with prolonged symptoms after SARS-CoV-2 infection in a large cohort of nearly 7000 participants in England [6].

The aim of this study was therefore to describe the persistent symptoms after SARS-CoV-2 infection in mainly non-hospitalized persons and any association with comorbidities or medications prior to the infection in the Lithuanian population during the pandemic’s first waves in 2020–2021. The hypothesis was that comorbidities and daily medication prior to infection are associated with symptomatology during acute infection as well as during the period after infection.

## 2. Materials and Methods

Participants were invited to participate in the study by anonymously answering an internet-based questionnaire created with Google Drive. The questionnaire was distributed in the Lithuanian language through Lithuanian websites, including private/public Facebook groups, city/town/district hospitals, and in media outlets (Appendix A). The online survey encouraged participation independent of the presence or absence of persistent symptoms. The study was approved by the Kaunas Regional Ethics Committee for Biomedical Research (approval number: BE-2-65) and is registered at ClinicalTrials.gov ID: NCT05000229 (accessed on 11 August 2021). Materials concerning ethical permission, consent information given to participants, and the study protocol (slightly updated after the 1st survey) are available through a link to the questionnaire at the university’s website (https://lsmu.lt/en/about-lsmu/structure/medical-practice/faculty-of-nursing/projektine-veikla/, accessed on 11 August 2021). Questions used in the survey were designed to capture main sociodemographic data; data related to acute SARS-CoV-2 infection, including laboratory findings, treatment during acute infection, comorbidities, and daily use of medication prior to infection; persistent symptoms after the infection; and attitudes related to the need for rehabilitation. The questions were exploratory, and no validated questionnaire was used in the survey. Informed consent was obtained from all participants. Appendix A summarizes the survey of participants from the 9th of June to the 9th of August 2021. In this study, we present the first collection of data and do not include the questions regarding the need for rehabilitation. 

The following inclusion criteria were required for participating: (1) age 18 years or older, (2) known SARS-CoV-2 infection with or without specific diagnostic tests or laboratory findings (PCR, antigen test, antibodies), and (3) at least 28 days after acute infection when participating in the survey. 

The following exclusion criteria were applied: hospitalized patients still receiving treatment or rehabilitation after SARS-CoV-2 infection, unstable or untreated comorbidities, or ongoing stabilization of comorbidities. None of the participants indicated the presence of exclusion criteria in the questionnaire.

Regarding comorbidities, the participants were asked to indicate the presence of any chronic disease prior to SARS-CoV-2 infection. Furthermore, the comorbidities were selected from preselected choices for 21 disorders along with an optional choice to write in other diseases. Two of the authors (DV and IBL) analysed and grouped the comorbidities because participants could indicate comorbidities either as preselected choices and/or describe them by name as an optional choice. Analysis of all answers resulted in 49 comorbidities, and all reported comorbidities were included in the statistical analysis.

Regarding acute and persistent symptoms, there were 18 preselected acute symptoms and 62 preselected persistent symptoms as well as the possibility to leave additional symptoms or comments. 

Medication was asked about as the presence or absence of “daily” medication before SARS-CoV-2 infection. Medication for acute SARS-CoV-2 infection was also asked about to understand if there was a difference between the studied groups during the acute course. 

### Statistics

Data were downloaded into Microsoft Excel and further analysed using SPSS version 27. Nominal data were compared using two-tailed Chi-square tests and are presented as numbers/percentages. Symptoms were also re-calculated as the total number of acute and persistent symptoms. Spearman correlation was used to investigate the associations between the persistent symptoms and the following parameters of interest: chronic disease at the time of acute infection, taking daily medication prior to acute infection, and acute symptoms during infection. Spearman correlation analysis was also performed between taking any daily medication and reporting any comorbidity prior to acute infection. The predictors of persistent symptoms (dependent variable) were analysed with a binary logistic univariate regression model regarding the following covariates: the presence/absence of chronic disease and medication prior to SARS-CoV-2 infection and the total number of acute symptoms. Gender and age were included as control variables. Statistical significance was considered as *p* < 0.05. 

## 3. Results

### 3.1. Study Cohort

The final study cohort included 1050 participants. The mean age was 41 years (SD 12 years, range 18–80 years), and 95% of the participants were younger than 60 years. Among the total cohort, 927 (88%) were women and 123 (12%) were men (Table 1). A major portion of the participants were from the biggest cities in Lithuania, including Kaunas (the second-largest city in Lithuania, 33%), Vilnius (the capital of Lithuania, 24%), and Panevėžys, 13%), and a total of 77% were living in an urban environment. Laboratory testing was performed for 965 (92%) participants, and asymptomatic infection (no acute symptom chosen) was found for 30 (3%) participants. The vast majority, 999 (95%), stayed at home during the course of their infection, while 51 (5%) were treated in the hospital, and most participants, 975 (93%), were working at the time of the survey. The mean time span between start of acute SARS-CoV-2 infection and completing the questionnaire was 25 weeks (SD 10 weeks, n = 1050). 

### 3.2. Comorbidities and Medication Prior to SARS-CoV-2 Infection

Among the 1050 participants, 538 (51%) reported not having any chronic disease and 762 (73%) reported not taking any daily medications prior to their SARS-CoV-2 infection. Among those reporting comorbidities (n = 512), only 31% (n = 157) were taking daily medication prior to the SARS-CoV-2 infection, which was higher compared to healthy participants taking medication (n = 131 among 538, 24%) (*p* = 0.023, Chi-square 2-tailed test). A weak but statistically significant correlation was found between taking daily medication prior to acute infection and having a chronic disease (r_s_ = 0.07, *p* < 0.022, N = 1050). 

Analysis of comorbidity data (the presence of chronic disease prior to SARS-CoV-2 infection) revealed 49 comorbidities (Table 2 and Appendix A). The total number of comorbidities varied from 0 to 9 in the cohort, with a median of 0 comorbidities. More detailed information on comorbidities is presented in Appendix A. Comorbidities were grouped according to major disease groups as follows: cardiovascular (19%); endocrine, excluding thyroidea (12%); thyroidea-related (8%); neurological (10%); psychiatric (7%); gastrointestinal (7%); allergic (7%); pulmonary, including diseases of the upper respiratory tract (5%); inflammatory/rheumatic (3%); chronic pain syndrome (3%); haematological (2%); oncological (2%); dermatological (2%); renal (1.%); gynaecological (0.9%); immunodeficiencies (0.6%); and others (undefined autoimmune disorder 0.1%, glaucoma 0.1%, and otosclerosis 0.1%) (Table 2). Of those having comorbidities, the greatest portion had 1 comorbidity (50% of 512), while only 30 (6% of 512) participants reported more than 4 comorbidities among all disorders. 

### 3.3. Associations between Comorbidities, Daily Medications, and Acute Symptoms

Regarding acute infection, 21 (2%) participants reported becoming “officially” healthy from acute infection within 10 days, 625 (60%) within 2 weeks, 305 (29%) within 4 weeks, and 99 (9%) longer than 4 weeks. This means that a primary health care doctor terminated isolation and granted the participant´s return to work and/or social activities. Therefore, most participants (91%) officially recovered from their acute symptoms within 4 weeks. The numbers of participants who did not receive any treatment, those who consumed prescribed medication, and those who treated themselves (not being prescribed any specific medication) during the acute SARS-CoV-2 period were approximately the same at 277 (27%), 403 (38%), and 370 (35%), respectively. 

Regarding the development of symptoms during the acute infection, half of the participants (51%) reported variation in their symptoms, while the remaining participants reported that their symptoms continued to be the same (26%) or that they developed new symptoms (18%). The total number of acute symptoms ranged from 0 to 18 with a median of 6 symptoms. The frequency of acute symptoms is presented in Figure 1A, showing the typical symptoms for SARS-CoV-2. The most frequent symptoms during acute infection were fatigue (765 participants (73%)), altered sense of smell (750 participants (71%)), fever/chills (679 participants (65.6%)), muscle/bone pain (633 participants (60%)), altered sense of taste (566 participants (54%)), runny nose (438 participants (42%)), sore throat (391 participants (38%)), dry cough (379 participants (36%)), shortness of breath (388 participants (32%)), lower back pain (340 participants (32%)), anxiety (328 participants (31%)), and insomnia (325 participants (31%)), while other symptoms were reported by fewer than 20% of the study cohort. Participants indicating the presence of comorbidities prior to their SARS-CoV-2 infection reported dry cough significantly more often during the acute infection (Figure 1A). Participants reporting medication prior to their SARS-CoV-2 infection reported fatigue, fever/chills, cough with mucus production, stool problems, and nausea significantly more often during the acute infection (Figure 1B). No significant correlation was found between taking daily medication prior to acute infection and the presence of any acute symptom (r_s_ = 0.02, *p* < 0.61, N = 1050).

### 3.4. Associations between Comorbidities, Medication, and Persistent Symptoms

Persistent symptoms were reported by 812 (77%) participants. The total number of persisting symptoms ranged from 0 to 48 with a median of 4 symptoms. Figure 2 shows the 38 symptoms reported by 5% or more of the study cohort. All 62 symptoms are presented in Appendix A. Reduced physical capacity was the most frequent persistent symptom, reported by 423 participants (40%), and this was followed by fatigue (405 participants (39%)), memory problems (354 participants (34%)), concentration problems (313 participants (30%)), altered sense of smell (247 participants (24%)), headache (209 participants (20%)), tachycardia (207 participants (20%)), unstable mood (199 participants (19%)), hair loss (183 participants (17%)), insomnia (181 participants (17%)), shortness of breath (158 participants (15%)), migrating body pain (154 participants (15%)), anxiety (149 participants (14%)), dizziness (140 participants (13%)), altered sense of taste (126 participants (12%)), increased sleepiness (125 participants (12%)), and night sweats (111 participants (11%)), while the remaining symptoms were reported by fewer than 11% of the study cohort. When comparing the persistent symptoms, participants who reported being unhealthy before their SARS-CoV-2 infection more often reported fatigue (Figure 2A). Participants reporting daily medication prior to their SARS-CoV-2 infection more often indicated reduced physical capacity, fatigue, memory problems, headache, tachycardia, hair loss, shortness of breath, migrating body pain, dizziness, night sweats, muscle/joint/bone pain, sensation of chest pressure, increased body weight, blurred vision, increased blood pressure, low back pain, muscle cramps, stomach reflux, lump in throat, cold/numbness/pain in the feet and hands, fear, and thoracic /neck pain among the symptoms reported by at least 5% of the participants (Figure 2B). Altered sensations of smell and taste were reported more often in those not taking any medication prior to their SARS-CoV-2 infection (Figure 2B). Among all symptoms, 30 of 62 were indicated as being significantly different regarding the intake of daily medication (Appendix A). There was a weak but statistically significant correlation between taking daily medication prior to acute infection and persistent symptoms (r_s_ = 0.1, *p* < 0.001, N = 1050). A weak but statistically significant correlation was also found between acute and persistent symptoms (r_s_ = 0.21, *p* < 0.001, N = 1050).

### 3.5. Regression Analysis for Predictors of Persistent Symptoms 

The predictors of persistent symptoms were analysed with a binary logistic regression model using the presence/absence of persistent symptoms as the dependent variable and the presence of daily medication prior to SARS-CoV-2 infection, the total number of comorbidities, and the total number of acute symptoms as the covariates. Gender and age were included as control variables. Table 3 summarizes the unadjusted and adjusted regression coefficients, showing that daily medication before SARS-CoV-2 infection and the total number of acute symptoms were the major predictors for persistent symptoms. Those who reported taking any daily medication before their SARS-CoV-2 infection had increased odds up to 1.66 (95% CI 1.1–2.44) for developing persistent symptoms after the infection. The number of total acute symptoms was also a predictor for persistent symptoms with increased odds up to 1.40 (95% CI 1.32–1.48). 

## 4. Discussion

In the present study we identified 49 disorders prior to SARS-CoV-2 infection and 62 persistent symptoms after 4 weeks post-infection. However, not all participants with chronic disorders reported taking daily medication prior to the infection. This was further confirmed by a weak correlation between the presence of comorbidities and taking daily medication before SARS-CoV-2 infection. We report here that those participants who took daily medication prior to their SARS-CoV-2 infection more often reported both acute and persistent symptoms, which is a key finding of this study. Among the persistent symptoms; fatigue; pain; and symptoms related to the nervous system, including the autonomic nervous system; as well as gastrointestinal/metabolic disturbances were reported more often. The fact that not all 62 symptoms were more often reported by participants taking daily medication prior to the infection might explain a weak but significant correlation coefficient between these parameters. 

The study cohort involved persons with milder forms of SARS-CoV-2 infection who mainly stayed at home during the acute infection instead of going to the hospital and who mainly were well-educated working women. The majority of participants were working at the time of the survey, indicating that despite persistent symptoms there was a restoration of working capacity. This confirms that a relatively healthy population was involved in the study. Similar findings regarding age, gender, education, and employment were obtained in a Swedish study recruiting persons with persistent post-COVID-19 condition for an online rehabilitation programme [14]. 

The lack of correlation between comorbidities and daily medication before infection might be explained by a relatively healthy population in the present cohort and broadly accepted non-pharmacological treatments in Lithuania, such as alternative medicine and physical rehabilitation medicine. On the other hand, some participants reporting to be healthy prior to SARS-CoV-2 infection indicated pharmacological treatment, which might include temporary over-the-counter medication for general or milder conditions such as pain, allergic reactions, etc. Participants in the present study were recruited by social media (Facebook), including three websites dedicated to medical staff. Hypothetically, this might explain the particular findings of a weak correlation between medication and comorbidities in this cohort. 

In clinical practice, comorbidities are often considered as risk factors for further symptoms or new disorders. For SARS-CoV-2 infection, cardiovascular and endocrine comorbidities were reported to be associated with increased severity of infection in hospitalized patients [15]. In this cross-sectional study, cardiovascular and endocrine comorbidities dominated over the others, and both were found in approximately 20% of the participants. Comorbidities as predictors of post-COVID-19 condition in inpatients and outpatients have been previously hypothesised [11,12]. However, the results of the present study could not identify comorbidities being a risk factor for persistent symptoms. Studies comparing both comorbidities and medication before SARS-CoV-2 infection as risk factors for post-COVID-19 condition are lacking. Altered senses of smell and taste as persistent symptoms were found to be increased in those participants who did not take daily medication prior to their SARS-CoV-2 infection. Seropositivity regarding antibodies towards SARS-CoV-2 virus has been shown to be positively associated with ageusia/anosmia and fever in a non-hospitalized cohort [16], and thus one can hypothesize that medication might affect immune responses and therefore cause vulnerability to a broad spectrum of symptoms. On the other hand, comorbidities by themselves might not play an important role during mild SARS-CoV-2 infection, at least in a middle-aged population after a milder course of infection. However, our findings should be further investigated to identify the risk factors for development of post-COVID-19 condition and to improve the follow-up of patients at risk. Further studies should also investigate whether certain drugs or combinations of drugs are more predictive than others in the development of persistent symptoms. Recently, it has been reported that SARS-CoV-2 infection leads to increased burden on the health care system, including prescribed medications (bronchodilators, analgesics, anticoagulants, etc.) for breakthrough infections after vaccination [17], but no consideration has been given to the prescription of medicines prior to the infection. It might be hypothesized that treatment of acute symptoms with prescribed medication during a limited time might be important for persistent symptoms, including mild SARS-CoV-2 infection. Again, the careful follow-up of prescribed medication during the acute phase might be more important than clinicians are accustomed to in practice. On the other hand, medication for the management of persistent symptoms might hypothetically be considered an increased risk factor *per se* for remaining symptomatology, even if such medication is essential for the management of both severe acute and post-COVID-19 conditions. Careful follow-up of prescribed medication and restoration of impaired functions by rehabilitation instead of overflow medication might be important in preventing persistent symptoms and/or breakthrough infections after vaccination. 

Our results are in line with previous reports on post-infection symptoms after SARS-CoV-2 infection such as fatigue, cognitive symptoms [18,19,20,21], shortness of breath, and altered sense of smell and taste [22]. The results of this study confirm that the total number of acute symptoms predicts persistent symptoms, which is in line with a study reporting that having more than five symptoms during acute infection is associated with post-COVID-19 condition [13]. The remaining burden of post-COVID-19 condition was predicted by acute infection and poorer health before infection [17]. 

Age, BMI, and female gender have been shown to be predictors for post-COVID-19 condition using an online survey [13]. In previous studies, women dominated among persons with post-COVID-19 condition after mild infection, and being a woman was also associated with prognosis for persistent symptoms in both hospitalized [23] and non-hospitalized persons [13,24]. However, another study did not confirm this association [25]. Our findings suggest that despite overrepresentation of middle-aged women in this study cohort, the female gender by itself was not associated with more frequent presence of persistent symptoms. Therefore, a larger recruitment of male participants is of importance in future investigations. 

The following limitations should be considered: (1) our online survey with limited spread in social media mostly recruited middle-aged women and did not include other populations, i.e., people who do not use the internet or do not pay attention to or participate in online surveys, which limits generalizability; (2) the anonymity of our survey did not allow contacting the participants in case of uncertainties regarding the inclusion criteria or missing data; (3) our first collection of data did not include a vaccination question (mass vaccination in Lithuania started in May 2021), which hypothetically could be an important confounder; (4) more data regarding comorbidities, the use of medication are required, for example, ongoing, prescribed, and non-prescribed medication as well as information regarding time frame for taking or not taking any medication; and (5) we used a retrospective design based on self-reported data. Moreover, the present study was initiated and performed prior to the publication of the WHO´s definition of post-COVID-19 condition and its report on persistent symptoms after SARS-CoV-2 infection [1]. Therefore, the present results are focused on persistent symptoms after SARS-CoV-2 infection rather than on post-COVID-19 condition itself. Due to missing data for inclusion criteria regarding the day of infection and duration of symptoms, approximately 16% of participants were excluded. However, further analysis with excluded persons did not change the results of the study.

The strengths of the present study are (1) the breadth of our questionnaire in regards to health status and medication before infection; (2) the inclusion of mostly middle-aged participants (40–49 years old), who statistically represent the age group most often infected with SARS-CoV-2 in Lithuania [26]; and (3) the expanded analysis of comorbidities when using both preselected and free answers.

## 5. Conclusions

The results of the present study indicate that daily medication before SARS-CoV-2 infection might be important for both acute and persistent symptomatology. This should be further studied in both hospitalized and non-hospitalized populations as well as for different drug groups, including non-prescribed drugs. Our findings also point out the importance of non-pharmacological strategies in the management of both comorbidities and persistent symptoms, for example, by endorsing exercise, psychotherapy, and rehabilitation interventions.

## Figures and Tables

**Figure 1 jcm-11-06278-f001:**
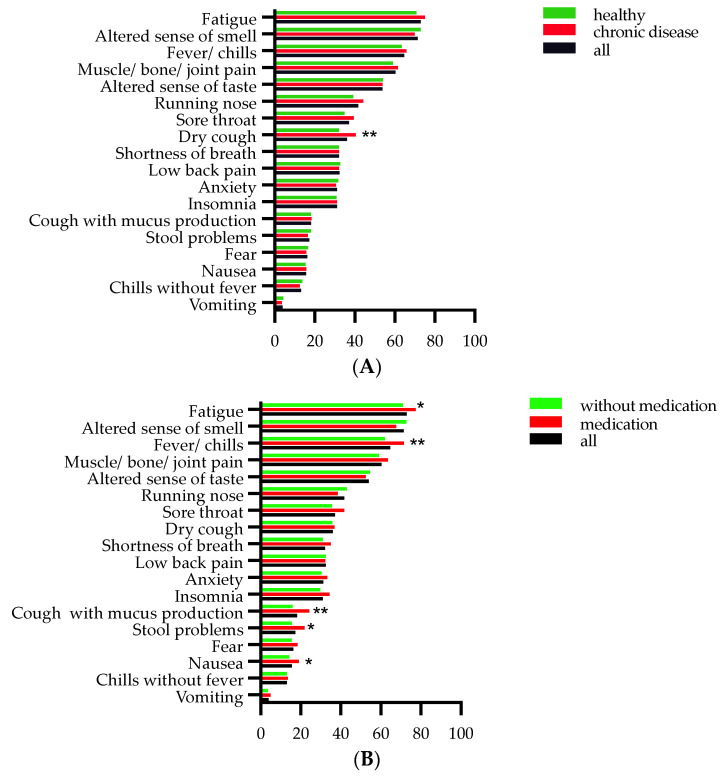
Acute symptoms in 1050 participants. Panel (**A**) presents percentages of healthy participants (green) and with chronic disease (red) reporting any acute symptoms prior to acute infection. Panel (**B**) presents percentages of participants without (green) and with medication (red) prior to acute infection. Statistically significant differences are indicated between healthy and with chronic disease (panel (**A**)) and between participants with and without medication (panel (**B**)) prior to acute infection. * *p* < 0.05 and ** *p* < 0.01 (Chi-square test).

**Figure 2 jcm-11-06278-f002:**
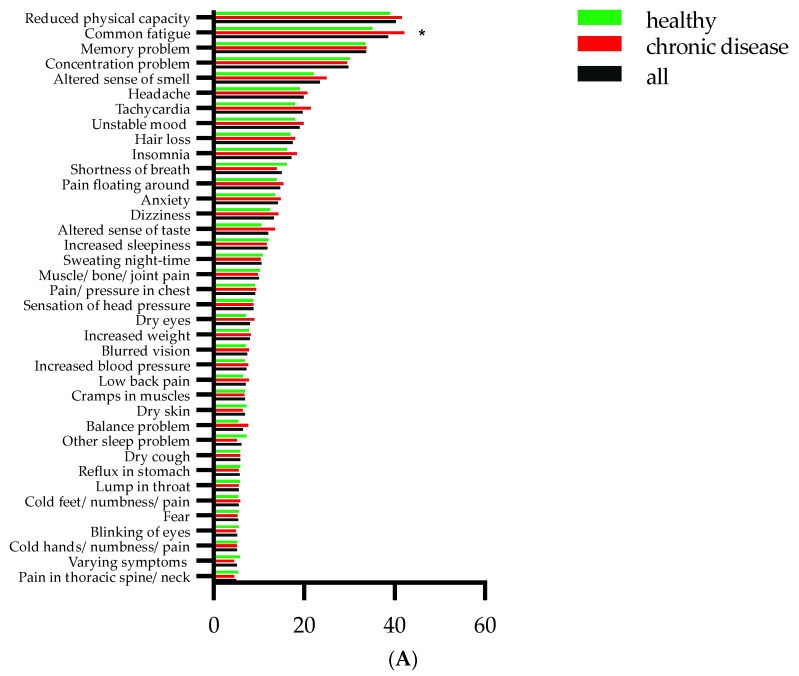
Persistent symptoms in 1050 participants. Panel (**A**) presents percentages of healthy participants (green) and with chronic disease (red) reporting any persistent symptom. Panel (**B**) presents percentages of participants without (green) and with medication (red) reporting any persistent symptom. Statistically significant differences are indicated between healthy and with chronic disease (panel (**A**)) and between participants with and without medication (panel (**B**)) prior to acute infection. * *p* < 0.05, ** *p* < 0.01, and *** *p* < 0.001 (Chi-square test).

**Table 1 jcm-11-06278-t001:** Socioeconomic characteristics of participants are presented as numbers and percentages in the total cohort and in both subgroups (with and without persistent symptoms).

		Total, n = 1050	With Persistent Symptoms n = 812	Without Persistent Symptoms n = 238
**Gender**	Female	927 (88.3%)	719 (88.5%)	208 (87.4%)
	Male	123 (11.7%)	93 (11.5%)	30 (12.6%)
**Age group**	Younger than 40 years	520 (49.5%)	405 (49.9%)	115 (48.3%)
	41–60 years	475 (45.2%)	370 (45.6%)	105 (44.1%)
	61–80 years	55 (5.2%)	37 (4.6%)	18 (7.6%)
**Education**	Primary/secondary	134 (12.8%)	102 (12.5%)	32 (13.5%)
	Higher non-university	327 (31.1)	252 (31%)	75 (31.5%)
	Higher university	589 (56.1%)	458 (56.4%)	131 (55%)
**Socioeconomic situation**	Employed/working	975 (92.9%)	755 (93%)	220 (92.4%)
	Temporary unemployed	23 (2.2%)	17 (2.1%)	6 (2.5%)
	Unemployed	42 (4%)	34 (4.2%)	8 (3.4%)
	Retired	9 (0.9%)	5 (0.6%)	4 (1.7%)
	Student	1 (0.1%)	1 (0.1%)	0

**Table 2 jcm-11-06278-t002:** Disorders presented as the number of participants and percentage of the whole population (in brackets). The total population was 1050 participants.

Disorders	Number of Participants	The Most Common Condition within Each Group
Cardiovascular	194 (19%)	High blood pressure, n = 153 (15%)
Endocrine	121 (12%)	Obesity, n = 96 (9%)
Thyroidea-related	82 (8%)	Hypothyroidism, n = 53 (5%)
Neurological	103 (10%)	Unspecified neurological diseases, excluding sleep disorders and epilepsy, n = 65 (6%)
Psychiatric	74 (7%)	Anxiety, n = 55 (5%)
Gastrointestinal	74 (7%)	Unspecified diseases of gastrointestinal tracts, n = 69 (7%)
Allergies	71 (7%)	Unspecified allergic diseases, n = 71 (7%)
Pulmonary	53 (5%)	Asthma, n = 34 (3%)
Inflammatory rheumatic	34 (3%)	Unspecified rheumatic diseases, n = 33 (3%)
Chronic pain	32 (3%)	Chronic pain syndrome, n = 25 (2%)
Haematological	19 (2%)	Anaemia, n = 12 (1%)
Oncological	18 (2%)	Unspecified oncological diseases, n = 18 (2%)
Dermatological	18 (2%)	Unspecified skin diseases, n = 12 (1%)
Renal	11 (1%)	Unspecified kidney diseases, n = 11 (1%)
Gynaecological	10 (0.9%)	Unspecified gynaecological diseases, n = 8 (0.8%)
Immunodeficiency	6 (0.6%)	Unspecified immunodeficiency diseases, n = 6 (0.6%)
Others:	3 (0.3%)	Other 3 (0.3%)

**Table 3 jcm-11-06278-t003:** Regressors predicting persistent symptoms after SARS-CoV-2 infection.

Regressors	Crude OR (95% CI), *p*-Value	Adjusted OR (95% CI), *p*-Value
Medication before infection	1.83 (1.28–2.62), <0.001	1.66 (1.10–2.44), 0.009
Total comorbidities	1.00 (0.9–1.11), 0.99	0.98 (0.9–1.1), 0.77
Total number of acute symptoms	1.40 (1.32–1.49), <0.001	1.40 (1.32–1.48), < 0.001
Gender	0.9 (0.58–1.39), 0.63	0.89 (0.55–1.43), 0.6
Age	1.0 (0.99–1.0), 0.75	1.0 (0.99–1.0), 0.9

## Data Availability

The data that support the findings of this study are available from the first author (D.V.) upon reasonable request.

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
