# Peer review of "Do Comorbidities and Daily Medication before SARS-CoV-2 Infection Play a Role in Self-Reported Post-Infection Symptoms?"

_jcm, 2022, doi:10.3390/jcm11216278_

Round 1
Reviewer 1 Report
1. The main strength of the study is the large amount of patient included; however, the choice of a social network for marketing represents a considerable selection bias, and this fact should be underlined. Moreover, it is unclear if the tool used for the data collection is a validated questionnaire.
2. Use of chronic medications could represent itself the cause of exhacerbation of symptoms of long COVID, i.e. if anti-hypertensive therapy causes hypotension afeter the infection due to the concomitance of dysautonomic disorders. Also this topic should be underlined, even in the abstract, because there is no apparent link with frailty.
Reviewer 2 Report
The manuscript entitled: “Do comorbidities and medication before SARS-CoV-2 infection play a role for self-reported post-infection symptoms?” is a study on the association of health status prior to acute infection with SAES-CoV2 and post-COVID conditions based on data of self-reported online surveys. The authors suggest that reported daily intake of medications prior to infection predicted persistent symptoms post-COVID.
In this study persistent post-COVID symptoms were reported by 812 of 1050 individuals in Lithuania during the period June 2021 – August 2021 in online surveys posted on the Kauno klinikos homepage and on eight different facebook pages. The majority of the recruited individuals were middle aged working women. The survey that was conducted between 5 and 45 weeks after the acute infection. Strengths of the study include a broad questionnaire that included open questions. Limitations of the study are the homogeneity of the study participant, restricting the generalizability. Also, the low correlation between daily medication and chronic disease prior to acute SARS-CoV2 infection is unexpected and affect the overall impression of the study.
I have the following questions:
Major
1) The questions of the survey are not available in the manuscript. The type of questions should be described in the methods, since the survey was in Lithuanian. For instance, the comorbidities and symptoms listed in the survey should be available in the manuscript, as well as examples of common non-listed (free text) symptoms/comorbidities. Also, I don't really understand the rationale to include a number of non-reported diseases in table S3? Can you explain your thinking?
2) I find table 1 hard to read the way the percentages add up in each sections. Can you please try to reorganize and improve readability?
3) Can you perform a Spearman correlation on binominal variables such as “having a chronic condition” and “daily medication”? The low correlation of “chronic disorder” and “taking daily medication” prior acute infection, is a bit peculiar. The authors address this in the discussion, but I still find the result odd.
4) Can Table 3 be clarified? Is the correct interpretation that daily medication before infection increased the risk of post-COVID conditions 1.66 times? Is it possible to change Exp(B) to a more intuitive heading?
5) Participants on medication prior to infection reported more acute symptoms than those not on medication (Fig. 2b). This causes reasons to think that the two explanatory variables Medication prior to infection and Total number of acute symptoms co-vary. How does prior medication correlate with persistent symptoms alone, and how does prior medication correlate with acute symptoms? And, does persistent symptoms correlate with acute symptoms? I think there is a risk that the authors overestimate the predictive power of prior medication for persistent symptoms. One potential remedy is to include one or more control variables to solidify your regression model.
6) Authors should also consider to lower the limit for statistical significance due to multiple testing, or at least discuss why p<0.05 was considered significant, when approximately 40 different symptoms are compared (Fig 2).
7) The paper will benefit from a thorough review of the English language.
Minor
1. There is something off with the table in Supplementary material 2.
2. How many individuals reported more than one comorbidity? More than 2? More than 3? In Table S3, It is hard to understand whether Indicating 2 and Indicating 3 refer to comorbidities in the same group?
3. Table S3: What is Zilber’s syndrome? Should it read Gilbert’s syndrome?
4. I miss information in the figures on how comparison been made.
5. Can you add the most common condition of each group in table 2?
6. Was there any difference in reported acute symptoms depending on how participants were diagnosed (with or without specific laboratory findings)?
7. Line 291-292: …and both were found in approximately 18% of the participants. - Did 18% of the participants have both CVD and endocrine disorder. Please clarify!
8. Include a discussion of the risk of bias in a study with a retrospective design based on self-reported data.
9. Line 349 typo: inclussion
Round 2
Reviewer 2 Report
I thank the authors for the clarifications and editing that has been performed, the authors have made adequate changes to the text. I only have two remaining minor concerns:
1. Supplementary material 2. The figure spans over two pages and I think that information still is missing. Please check!
2. Figure legends. I think it should be further clarified that statistical analyses compared. For example (if this was the case) participants with chronic disease and healthy participants (fig 1A), with and without medication (fig 1B), participants with and without chronic disease (fig 2A) and participants with and without reported medication (fig 2B).
Author Response
Dear Reviewer,
Thanks for your notices and apologise for my blindness. Presentation of figure in Supplementary Material 2 and figure legends are improved, which certainly increases readability. Therefore, I hope for positive final decision.
With best regards, Indre Bileviciute-Ljungar.
